# Directional and Weighted Urban Network Analysis in the Chengdu-Chongqing Economic Circle from the Perspective of New Media Information Flow

Changwei Xiao [1,†], Chunxia Liu [1,†] and Yuechen Li [2,3,*]

1 School of Geography and Tourism, Chongqing Normal University, 37 Daxuechengzhong Road, Shapingba District, Chongqing 401331, China
2 Chongqing Jinfo Mountain Karst Ecosystem National Observation and Research Station, Chongqing Engineering Research Center for Remote Sensing Big Data Application, School of Geographical Sciences, Southwest University, Chongqing 400715, China
3 Key Laboratory of Monitoring, Evaluation and Early Warning of Territorial Spatial Planning Implementation, Ministry of Natural Resources, Chongqing 401147, China
* Correspondence: liyuechen@swu.edu.cn
† These authors contributed equally to this work.

**Abstract:** The study of the two-way information flow between cities is of great significance to promote regional coordinated development, but the current mainstream non-directional network analysis method cannot analyze it effectively. In this paper, the quantities of relevant media articles in WeChat and Weibo between cities are taken as the traffic indices to construct a directional and weighted urban network of the Chengdu-Chongqing Economic Circle in China. Based on this network construction method, which adds direction thinking, we analyze the characteristics of information interconnection between cities. According to the analysis, we find that the provincial boundary hinders information interconnection, and the imbalance of external information interconnection is more serious in Chongqing's central urban area, Liangping, Ya'an and Mianyang. In addition, we analyze the centrality status of different cities in the outward and inward perspective and further explore the factors that cause these differences in centrality. The results show that the centrality of the information network is not sensitive to the basic strength of the city, and it is the accessibility, including high-speed rail transportation access and telecommunication access, which controls the centrality of the city network.

**Keywords:** information flow; urban network; centrality analysis; QAP analysis; Chengdu-Chongqing Economic Circle





## 1. Introduction

The characteristics of cooperation among cities have become diverse, with frequent and complex transfers of capital, organization, goods, people and other elements between cities. Multi-level and multi-type association networks are gradually formed in urban agglomerations. Scholars' research on the urban network started from the perspective of a topological network and is now gradually focusing on the flow space network [1].

For hierarchical topology networks [2–6], scholars generally assign the right to the city based on the vertical branch level of the company organization and quantify the agglomeration and diffusion ability, centrality and status of each city. However, scholars use these kinds of data to analyze the urban network structure, which emphasizes the horizontal relationship and cannot take the number of connections between cities into account. Due to the limitation of data accessibility, it is difficult to obtain the vertical branch structure of micro and small organizations. This lack of data means that the hierarchical topology method tends to only include data from medium and large organizations and mostly focuses on high-grade cities as the research unit. The flow space network is derived from

the flow space theory [7]. According to this theory, flow space is a material organization of social practice with a shared time that operates through element flow. The emergence of the flow space theory provides a dynamic analysis perspective for urban network research and constructs a new spatial dynamic mechanism [8]. Due to reductions in the difficulty of data acquisition of flow factors [9], logistics [10–12], population flow [13–19] and information flow [20–23] have received more and more attention.

Information flow reflects the transfer of economic, social, cultural and other factors, and it is a necessary supplement to the physical flow space [24]. Compared with other flow networks, the constitutive elements of information flow are less affected by physical space and time. Therefore, information flow is more representative and suitable for analyzing the level of inter-city connection. At present, the research of information connection is mainly based on Google, Baidu and other search engines [20,25,26]. However, there is a lack of research on the information connection between new media platforms in cities. Mainstream urban media platforms, such as WeChat and Weibo, regularly release articles containing economic, social, cultural and other information about various cities, which are highly comprehensive [27–29]. Meanwhile, most of the existing information flow studies are based on the analysis of undirectional networks, and this research uses the total number of links between two cities as the flow of information flow. The advantage of the approach is that it can concisely represent the topological structure and node status characteristics of the city network, but it fails to quantify the asymmetry of two-way information flow between cities because even the seemingly symmetrical city linkage relationship may harbor great asymmetry. Effectively measuring and visualizing the asymmetry of flows in directional weighted networks and quantifying the node characteristics exhibited by such asymmetric links have become a new focus of urban network research [30].

As the connecting point of "the Belt and Road Initiative" and China's Yangtze River Economic Belt, the Chengdu-Chongqing Economic Circle has a high potential for development, with a dense population, numerous cities and diverse scenery [31–33]. However, the cities in the economic circle have similar factor endowments, are all at similar stages of economic development and have low levels of collaborative development [34]. Studying the characteristics of information connection between cities can effectively judge the pattern of the Chengdu-Chongqing Economic Circle and the level of inter-city relations. Previous research works about the urban network in the Chengdu-Chongqing Economic Circle mainly focus on the analysis of network structure and development [35–38]. These studies confirm the core status of Chengdu and Chongqing and show that the urban agglomeration has a hub-and-spoke organization mode, which evolves into the whole area network. However, the asymmetric connection between cities in the Chengdu-Chongqing Economic Circle has not been analyzed because the directionality of flow has not been sub-divided.

In this paper, we analyze the information flow network in the Chengdu-Chongqing Economic Circle according to the directional articles of inter-city media platforms. We use the interconnection imbalance analysis and centrality analysis to analyze the network from three levels of one-way flow, interconnection and urban nodes and compare the influencing factors with quadratic assignment procedure regression analysis. Overall, this study provides the necessary theoretical and technical information to support the high-quality, coordinated development of the Chengdu-Chongqing Economic Circle.

## 2. Overview of the Study Area and Data Sources

### 2.1. The Study Area

The Chengdu-Chongqing Economic Circle is located in the southwest of China and has a pleasant climate, rich products, large population and dense towns. It is an important connection point between the Belt and Road Initiative and China's Yangtze River Economic Belt, and it has good regional development advantages. The region has a total area of 185,000 square kilometers, covering 29 districts and counties in Chongqing, 20 districts, cities and counties in Chengdu and 14 sub-cities. In 2019, the permanent population of the region was 96 million, with a GDP of CNY 6.3 trillion. In this paper, the central urban area

of Chongqing (including Yubei, Yuzhong, Shapingba, Beibei, Dadukou, Jiangbei, Jiulongpo, Banan, Nanan) and the central urban area of Chengdu (including Jinjiang, Qingyang, Jinniu, Wuhou, Chenghua, Longquanyi, Qingbaijiang, Xindu, Wenjiang, Shuangliu, Pidu) are regarded as two cities, among a total of forty-five cities in the research area (Figure 1).

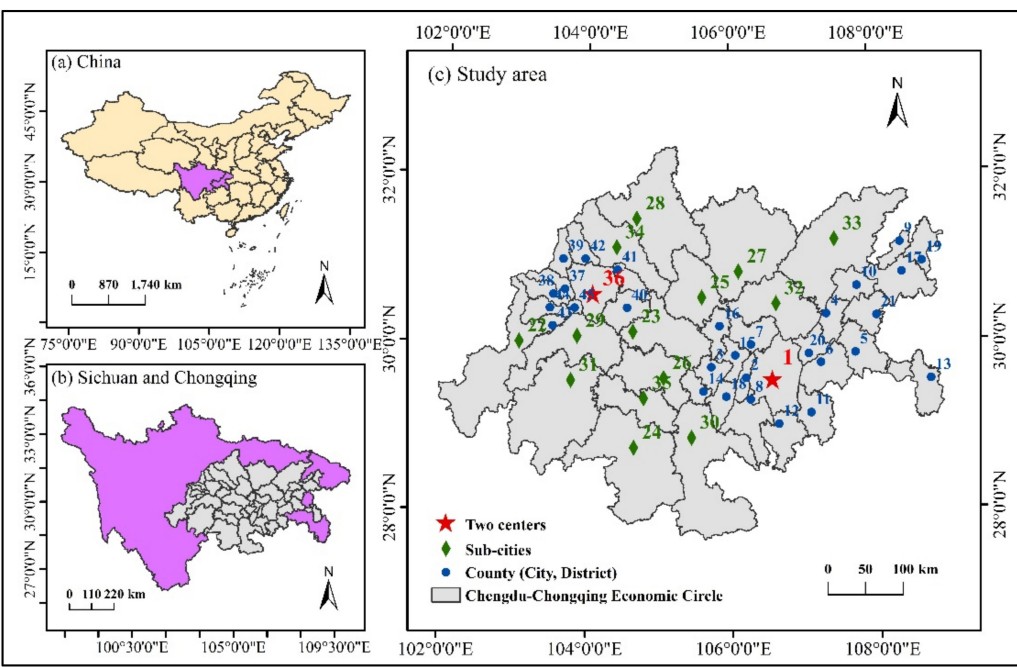

**Figure 1.** The study area. (**a**) Map of China, (**b**) map of Sichuan and Chongqing, (**c**) map of study area: 1 is the central urban area of Chongqing; 2 is Bishan; 3 is Dazu; 4 is Dianjiang; 5 is Fengdu; 6 is Fuling; 7 is Hechuan; 8 is Jiangjin; 9 is Kaizhou; 10 is Liangping; 11 is Nanchuan; 12 is Qijiang; 13 is Qianjiang; 14 is Rongchang; 15 is Tongliang; 16 is Tongnan; 17 is Wanzhou; 18 is Yongchuan; 19 is Yunyang; 20 is Changshou; 21 is Zhongxian; 22 is Ya'an; 23 is Ziyang; 24 is Yibin; 25 is Suining; 26 is Neijiang; 27 is Nanchong; 28 is Mianyang; 29 is Meishan; 30 is Luzhou; 31 is Leshan; 32 is Guang'an; 33 is Dazhou; 34 is Deyang; 35 is Zigong; 36 is the central urban area of Chengdu; 37 is Chongzhou; 38 is Dayi; 39 is Dujiangyan; 40 is Jianyang; 41 is Jintang; 42 is Pengzhou; 43 is Pujiang; 44 is Qionglai; 45 is Xinjin.

## 2.2. Data Sources

Wechat and Weibo have substantial user groups and a wide range of media information dissemination. Based on the number of monthly active users data released by the official websites of Tencent (http://tencent.com/ accessed on 16 May 2021) and Sina Weibo (http://ir.weibo.com/ accessed on 20 May 2021), in each quarter of 2018 and 2019, the average monthly active users of Wechat and Weibo reached 1.105 billion and 464 million, respectively; these statistics have great representational power and accuracy in describing the inter-city media information connection. We select the WeChat and Weibo public accounts of each city government to conduct the statistics of related article data.

Different from most nouns, which are malleable and polysemic, Chinese city names cannot be extended or shortened and have very specific usage scenarios. People mainly use city names to represent cities or attach information to cities. Therefore, they cannot be replaced by any other vocabulary, and this specificity makes it the best entry point for mining information links between cities. At present, the method has been widely studied and applied by scholars in the field of information association [20,25,39–41]. Based on this idea, this paper uses the official media on Wechat and Weibo platforms to carry out data innovation. In the account of any city A on the two platforms, we selected a fixed time range and used the name of city B as the keyword to conduct an internal search. The number of articles related to city B existing in city A in a fixed time range was calculated as the rate of

flow from city B to city A. Considering the synonyms of city names, sub-regions, nicknames and other problems affecting the accuracy of the data, we checked and corrected every piece of relevant media information during and after the statistics, especially the relevant data of the city names that are prone to problems, so as to ensure the accuracy of the data. Finally, we counted the correlation number of inter-city articles in the Chengdu-Chongqing Economic Circle from 1 January 2018 to 31 December 2019 and summarized it into the matrix data of "Weibo interconnection" and "WeChat interconnection".

In the analysis of influencing factors, explanatory variable data were sourced from the Statistical Yearbook 2019 of Sichuan Province, Chongqing Municipality and Chengdu City, the website of China Railway Customer Service Center (http://12306.cn/ accessed on 6 March 2021) and the statistical bulletin of the national economic and social development of cities in the study area in 2018.

### 3. Methods

#### 3.1. Logical Framework

The main steps of this paper are shown in Figure 2. (1) We collect statistics on the correlation data of media articles between cities on WeChat and Weibo platforms and summarize them into a matrix in the form of 45 × 45. (2) We construct a directional and weighted urban network based on matrix data. (3) We use GIS spatial analysis based on Arcgis 10.5, link symmetry analysis and QAP analysis based on Ucient 6.0 to analyze the directional connection, interactive connection and urban centrality in the directional weighted network. (4) We discuss and summarize the research results from the above three aspects.

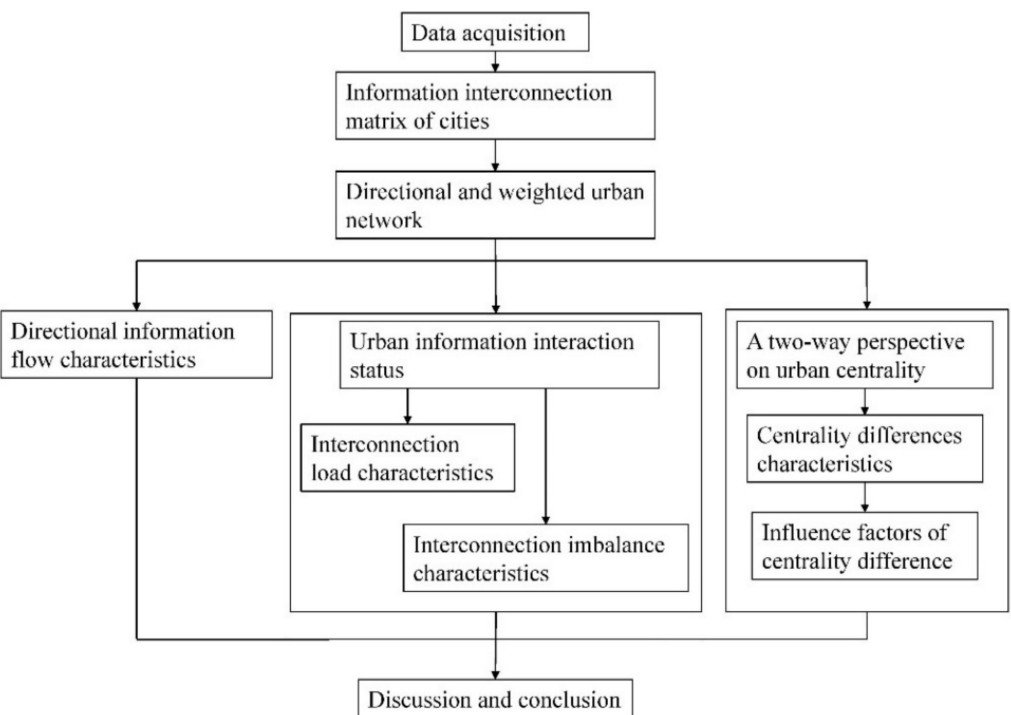

**Figure 2.** Logical framework diagram.

#### 3.2. Rate of Flow

The information diffusion level of each media platform is closely related to the number of monthly active users. Based on the average monthly live users of the two platforms in 2018–2019, we determined the weights of 0.7 and 0.3 for the two platforms, respectively. Finally, we weighted and summed the two platforms' data and obtained 2025 directional information flows for the research area.

### 3.3. Interconnection Features

3.3.1. The Interrelation Quantity

In the field of flow space, many studies have used the interrelation quantity of factor flow to analyze the total level of inter-city connections [15,19,23]. In this paper, the sum of new media information flow between two cities is defined as the interrelation quantity using the following calculation:

$$Q_{ij} = W_{ij} + W_{ji}, \tag{1}$$

where $Q_{ij}$ represents the interrelation quantity of new media information flow between city i and city j. $W_{ij}$ and $W_{ji}$ are bidirectional information flow between the two cities. The value range of $W_{ij}$ and $W_{ji}$ is $[0,+\infty)$. Considering that there is no bidirectional information connection between cities when one (or both) of $W_{ij}$ and $W_{ji}$ is 0, this paper determines that interconnection occurs only when both Wij and Wji are greater than 0.

3.3.2. The Degree of Interconnection Imbalance

Scholars in this field have succeeded in the research and have proposed and improved the formation of specific and repeatedly demonstrated analytical models [30,42]. The calculations are as follows:

$$F_{ij} = \frac{W_{ij}}{W_{ij} + W_{ji}}, \tag{2}$$

$$LSI_{ij} = 2F_{ij} - 1, \tag{3}$$

where $W_{ij}$ and $W_{ji}$ are the flow values of directional flow between cities i and j. The values of $W_{ij}$ and $W_{ji}$ are in the range $[0,+\infty)$. $F_{ij}$ refers to the ratio of the flow in the direction of i to j to the sum of flow between i and j. $LSI_{ij}$ is the symmetry degree of the link between city i and city j, with values in the range $[-1,1]$. When $LSI_{ij} = 0$, the two-way links are symmetrical. When $LSI_{ij} < 0$, the strength of the flow from i to j is smaller than that from j to i, and when $LSI_{ij}$ is $-1$, the interconnection is a one-way net flow from j to i. When $LSI_{ij} > 0$, the flow strength from i to j is greater than that from j to i, and when $LSI_{ij}$ is 1, the interconnection is a one-way net flow from i to j.

In this paper, the absolute value of link symmetry is used to evaluate the imbalance of the interconnection, while the imbalance mechanism is studied according to the comparison of bidirectional flow. The formula of the degree of interconnection imbalance is as follows.

$$UB_{ij} = \left| LSI_{ij} \right|, \tag{4}$$

where $UB_{ij}$ is the degree of interconnection imbalance between city i and city j, with a value range of $[0,1]$. The higher the value of $UB_{ij}$, the more significant the imbalance. $LSI_{ij}$ is the link symmetry degree between two cities, with reference to the analysis model of the scholars above [30,42].

### 3.4. Centrality Analysis

3.4.1. The Degree of Centrality

Centrality is an important research issue of social network analysis, which is used to analyze what power nodes have in the social network [43,44]. Considering that the information elements are different from traffic, goods, people and other elements, they are non-physical links between cities, and they complete the transmission process from any starting city to any receiving city at the moment of publication of Wechat and Weibo articles, which is extremely instantaneous and will not be restricted by any other city points. Therefore, it lacks significance and feasibility to analyze the related centrality, such as the betweenness centrality and closeness centrality of nodes in urban networks from the perspective of location, path distance and betweenness, so we only use the degree centrality method to measure the centrality. In an unweighted network, the degree of centrality characterizes the level of topological association between a city and the outside world. In a weighted network, the degree of centrality characterizes the specific amount

of connection between a city and the outside world. The weighted degree of centrality integrates the degree of topological association and the level of connection volume between the city and the outside world [45]. This approach provides a comprehensive analysis of both unweighted and weighted methods, but the difference between the outgoing and incoming perspectives cannot be identified. In this paper, we add directional thinking. We propose a method of combining the weighted degree of centrality with directionality to analyze the centrality of cities from a two-way perspective, rather than being limited to the centrality of urban nodes under a two-way total linkage perspective. These calculations are as follows:

$$DO_i = \frac{\sum_{j=1}^{n} A_{ij}}{n-1}; \ DI_i = \frac{\sum_{j=1}^{n} A_{ji}}{n-1}, \tag{5}$$

$$SO_i = \sum_{j=1}^{n} W_{ij}; \ SI_i = \sum_{j=1}^{n} W_{ji}, \tag{6}$$

$$WDOC_i = DO_i{}^{\alpha} \times (\frac{SO_i}{DO_i})^{(1-\alpha)}; \ WDIC_i = DI_i{}^{\alpha} \times (\frac{SI_i}{DI_i})^{(1-\alpha)} \tag{7}$$

where $A_{ij}$ and $A_{ji}$ denote the two-way topological connection between cities. A value of 1 indicates that an information flow connection exists, and a value of 0 shows that no connection exists. $W_{ij}$ and $W_{ji}$ denote the two-way information flow strength between cities. $DO_i$ and $DI_i$ denote the topological centrality of the outgoing and incoming directions of city i. $SO_i$ and $SI_i$ denote the node strength of the outgoing and incoming directions of city i. $WDOC_i$ and $WDIC_i$ denote the weighted out-degree and in-degree centrality of city i. $\alpha$ is an assigned parameter with a value of 0.5, following previous research [45].

3.4.2. QAP Regression Analysis

Quadratic assignment procedure (QAP) is a method for analyzing the relationships between relational matrices. QAP regression analysis is used to study the regression relationship between a single relational matrix and multiple other relational matrices to assess the significance of the coefficients of determination. These relationships are calculated by converging the values in the matrix grid into long vectors and performing multiple regression analysis and then randomly permuting the matrix ranks to calculate the regression coefficients [46]. The standard error is calculated by repeating this calculation several times.

This paper takes the difference of weighted out-degree centrality and the difference of weighted in-degree centrality between cities as dependent variables. Based on the goal of analyzing the key influencing factors of centrality difference, referring to the research experience of previous scholars, and considering the availability of data at the district and county level, in this paper, six representative independent variables are introduced from the population, economy [25,47], industry, capital [25,48], transportation and telecommunications [25]. The QAP regression analysis model is as follows:

$$\begin{aligned} F_i = d_0 + d_1 X_1 + d_2 X_2 + d_3 X_3 + d_4 X_4 + d_5 X_5 + d_6 X_6 \\ i = (1,2) \end{aligned} \tag{8}$$

where $F_1$ and $F_2$ are the weighted out-degree centrality difference matrix and the weighted in-degree centrality difference matrix, respectively; $d_1 \sim d_6$ are the regression coefficients; $d_0$ is a constant and $X_1 \sim X_6$ are the explanatory variables relationship matrices.

The specific selected explanatory variables are as follows. (1) People are an important carrier of urban association. This paper combines the data on the resident population in each city at the end of 2018 to construct the matrix of differences in the resident population between cities. (2) GDP per capita is an important indicator to measure the development of cities. This paper characterizes the economic level using data on the GDP per capita of each city in 2018 and constructs the matrix of GDP per capita differences between cities. (3) Compared with primary and secondary industries, the tertiary industry is more sensitive

to information. This paper combines data on the contribution of the tertiary industry in 2018 to construct the matrix of the difference in the contribution rate of the tertiary industry between cities. (4) Government public finance expenditure includes expenditure on various aspects, such as general public services, national defense and public security. This paper selects the public finance expenditure data of each city in 2018 to construct the difference matrix of public finance expenditure of each city. (5) Road network transportation is an important condition for cities to have factor exchanges with the outside world, and high-speed railroads are faster and have better transportation efficiency compared with general railways and roads. Considering that high-speed trains in China use D, G and C as the beginning letters of ticket numbers, in this paper, we obtain the number of train trips named D/G/C between city points from the 12306 website. These data are used to calculate the total number of trips between each city and other cities in the study area and to construct the matrix of the difference between cities. (6) WeChat and Weibo are closely related to the development of mobile internet and the possession of a cell phone. In this paper, we count the number of cell phone users per 100 people in each city and construct the inter-city cell phone users per 100 people difference matrix.

## 4. Results

### 4.1. Characteristics of Directional Information Flow

We divide the directional information flow into six levels based on the flow of the associated network, with a spacing of 50 units. A visual representation is shown in Figure 3, and this is used to study the characteristics of the layers in which different flows are located.

The central urban area of Chengdu is the core of the first and second levels of directional flow. The associated flows in the central urban area of Chengdu are higher than those in the central urban area of Chongqing. The first and second levels are dominated by the directional flows associated with Chengdu's central urban area, which is bidirectionally interconnected with the central urban area of Chongqing, Yibin, Dazhou and Nanchong in these two levels. The southern and northern cities in Sichuan have hierarchical differences in the directional connections with Chengdu's central urban area, with the associated southern flows being higher than the northern ones. The distribution within Chongqing is sparse, and only a few cities, such as Wanzhou, Liangping and Jiangjin, have directional links with the central urban area of Chongqing at these two levels.

The third and fourth tiers are dominated by information interconnection within Chongqing, and the central urban area of Chongqing is the core of these two tiers. The number of directional flows associated with Chongqing's central urban area increases significantly in the third and fourth levels. Most of the cities in southwest Chongqing have a large directional connection flow with the central urban area, which is concentrated in the third level. Among them, Rongchang, Tongnan and Qijiang are bidirectionally interconnected with Chongqing's central urban area for information. However, the directional links between the northeastern cities, such as Kaizhou, Yunyang and Zhongxian, and the central urban area of Chongqing are mostly at the fourth level, which is obviously weaker than cities in the southwest of Chongqing.

The fifth level is dominated by non-core linkages and cross-provincial linkages, and the majority of directional information flows are clustered in the sixth level. In the fifth level, the directional information flow is no longer limited to the connection between the two centers and other cities but primarily exists between non-core cities, such as Yibin, Leshan and Dazhou. The provincial boundary barriers are overcome at some key nodes at this level, and one-way as well as two-way flows of information across provinces emerge, such as between Dazhou and Chengkou, Luzhou and Chongqing central urban area, and Guang'an and Chongqing central urban area. Most of the remaining flows with directional information with values of less than 50 are clustered in the last level. These flows intertwine with each other to form a complex and dense network of correlated directional and weighted information flows.

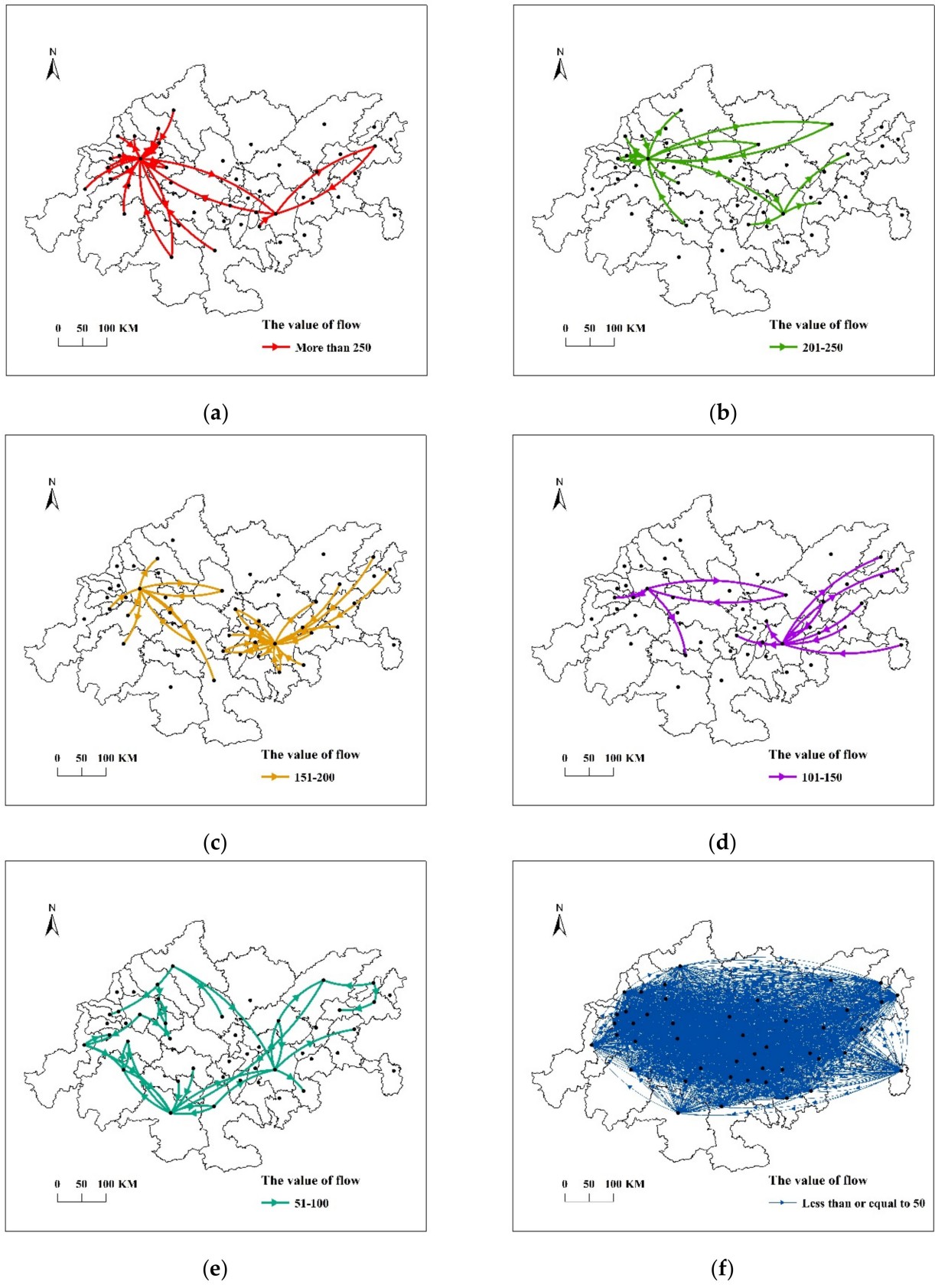

**Figure 3.** Six levels of directional information flow in study area. (**a**) Level 1, (**b**) Level 2, (**c**) Level 3, (**d**) Level 4, (**e**) Level 5, (**f**) Level 6.

Generally speaking, the central urban area of Chengdu and the central urban area of Chongqing, as regional economic, social and cultural centers, have a very high level of directional information connection in the urban network. However, in contrast, the level of information connections between the remaining cities in the region is significantly lower.

### 4.2. Analysis of the Interconnection of Information Flow

This paper uses the natural breakpoint method to divide the values of interrelation quantity into three tiers, and the degrees of interconnection imbalance are equally spaced into four tiers (Figure 4). Considering the lack of analytical significance of interconnection imbalance at low interrelation quantity and low imbalance degrees, this paper only conducts a two-way analysis of imbalanced interconnection with intermediate or higher interrelation quantity and imbalance degree value greater than 0.5 (Figure 5).

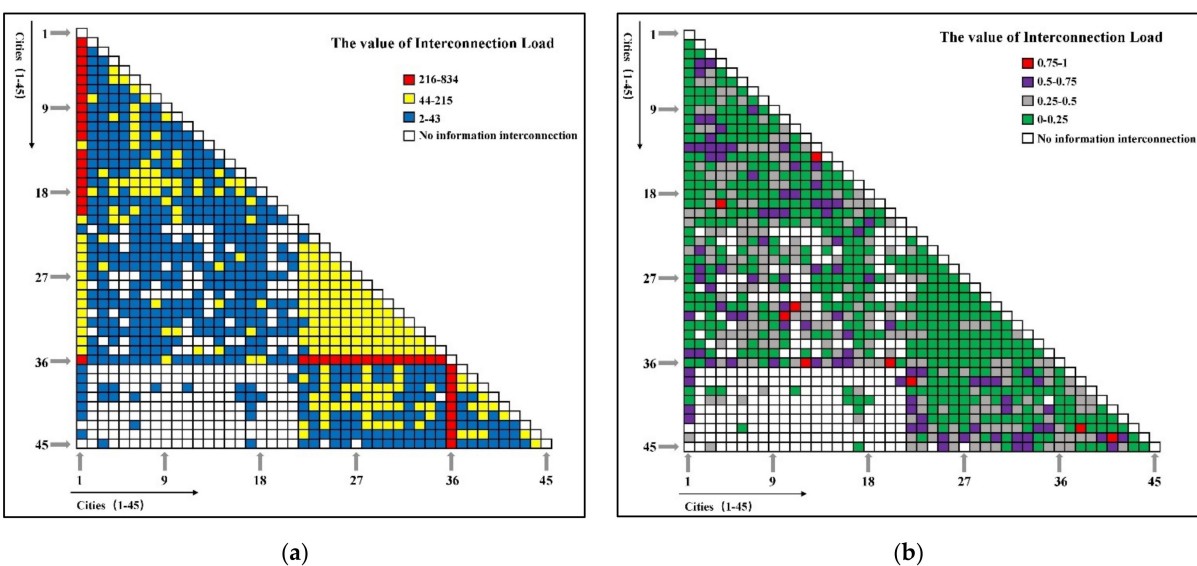

(a)                                                                                        (b)

**Figure 4.** Pattern of interconnection. (**a**) Pattern of interconnection interrelation quantity, (**b**) Pattern of interconnection imbalance. The cities corresponding to the numbers 1–45 are the same as in Figure 1.

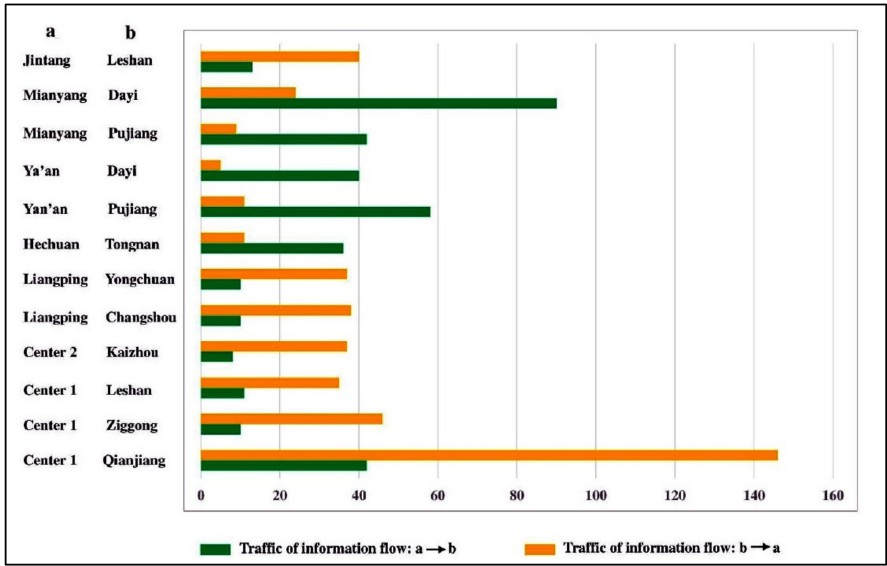

**Figure 5.** Two-way comparison of interconnections. Center 1 is the central urban area of Chongqing; Center 2 is the central urban area of Chengdu.

Firstly, the influence of provincial boundary on information interconnection is quite obvious. Between sub-cities and Chengdu's non-central districts and counties, there are many interconnections with interrelation quantity of more than 44. However, such interconnections are rare between sub-cities and Chongqing's non-central districts and counties, and we cannot even see the interconnections with sub-cities in some districts and counties. Meanwhile, the level of interconnection between Chengdu's non-central districts and counties and Chongqing's non-central districts and counties is even lower. Most of the interconnections between them do not occur, and even though a small part of the interconnections occur, their interrelation quantity is under 43.

Secondly, the two-way information flow between sub-cities is significantly coordinated. In the Chengdu-Chongqing Economic Circle, the external information interrelation quantity of sub-cities is second only to that of the two central urban areas. The information interrelation quantity between sub-cities and the twin centers and within sub-cities is all above the medium level. The third level is composed of information interconnection between sub-cities and non-central districts and counties. In the analysis of interconnection imbalance, there are more interconnections with imbalance values of 0.5–0.75 or even 0.75–1 within Chongqing and Chengdu, while the imbalance of information interconnection among sub-cities is mostly below 0.25, and only a few reach values from 0.25 to 0.5. The inter-sub-city information connection level is high and is able to maintain coordinated information interconnection and a high information interrelation quantity.

Thirdly, the imbalance of external information interconnection is serious in Chongqing's central urban area, Liangping, Ya'an and Mianyang. These cities associated with the central urban area of Chongqing, including Qianjiang, Zigong and Leshan, have interconnections that meet the medium-level or higher interrelation quantity. However, the information flows between these three cities and the center of Chongqing are not equal, with an imbalance greater than 0.5 and a large amount of information flows into the central urban area of Chongqing, but the reverse information flows are significantly lower. This result indicates that Qianjiang, Zigong and Leshan do not pay significant attention to Chongqing's central urban area and do not reciprocate the level of attention that Chongqing's central urban area pays to them. The imbalance mechanism of external information interconnection in Liangping is the same as that in Chongqing's central urban area, while Ya'an and Mianyang are obviously different. Compared with the outward disadvantage of Chongqing's central urban area, Ya'an and Mianyang show an inward disadvantage. The inflow of information from some cities is much lower than the outflow to other cities. The level of attention from outside cities is high, while Ya'an and Mianyang do not have the same level of reverse attention.

To sum up, the information interaction between cities in the Chengdu-Chongqing Economic Circle is severely constrained by administrative boundaries, and the level of information interaction between cities in Chongqing and Sichuan is relatively low. On the other hand, the information interaction between sub-cities is very coordinated. In obvious contrast, the information interaction between the central urban area of Chongqing, Liangping, Ya'an, Mianyang and other relevant cities is seriously unbalanced.

*4.3. Centrality Analysis*

4.3.1. Centrality Differences

Using the weighted out-degree centrality and weighted in-degree centrality of each city, this paper visualizes the centrality differences from a two-way perspective to study the spatial pattern across the region (Figure 6).

This analysis shows that the centrality of each city shows clear stratification in a bidirectionally information linkage network. Chengdu's central urban area and Chongqing's central urban area have the highest centrality and exhibit large differ-ences from second-tier cities such as Yibin, Leshan, and Mianyang. The information flow network of sub-cities shows obvious weakness in the east and strength in the west. The sub-cities are all within Sichuan province, but the centrality of the east and west in the information flow network

shows a large difference. Cities in the east, such as Guang'an, Nanchong, and Suining, mostly have lower weighted centrality, while cities in the west, such as Mianyang, Leshan, Yibin, and Luzhou, have significantly higher centrality. This difference between east and west forms a subdivision within the sec-ond tier. In addition, the central status of non central districts and counties in Chengdu and Chongqing is very low. This finding is especially true for Dayi, Qionglai, Qianjiang, Nanchuan, and Yunyang, which are not only at the overall bottom level in the out-ward perspective but are also significantly lower than other non-central districts and counties from the inward perspective.

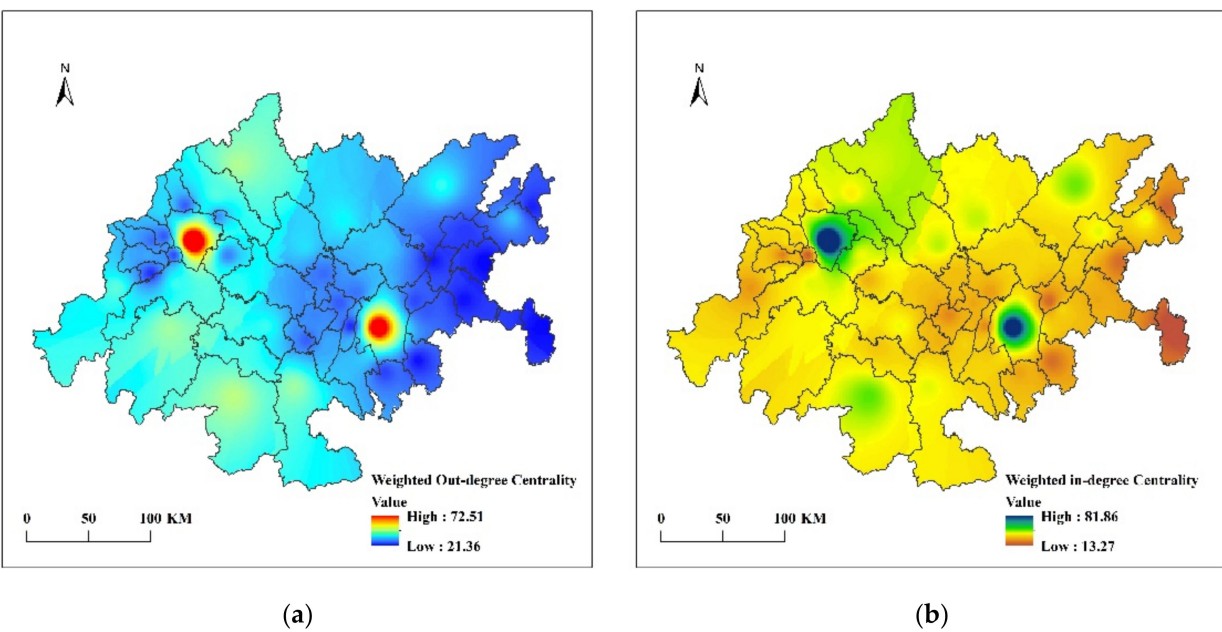

(**a**)                                                    (**b**)

**Figure 6.** Spatial pattern of centrality. (**a**) Spatial pattern of weighted out-degree centrality, (**b**) Spatial pattern of weighted in-degree centrality.

### 4.3.2. Influencing Factors of Centrality Differences

Two regression coefficients before and after standardization are calculated from the QAP analysis. Due to the large differences in the factor magnitudes of the explanatory variables, the standardized coefficients are selected for comparison in this paper (Table 1). The adjusted coefficients of determination from the two-way perspective are 0.931 and 0.865, which shows that the overall explanatory effect is good.

High-speed rail transportation has the most significant effect on dual-perspective centrality. The D/G/C train-shift count contributes to more than 60% of the difference between the weighted-out and weighted-in centrality, and both pass the 1% significance test. High-speed railroad construction is closely related to the centrality of an information network. Cities with high-speed railway extensions or high-speed railway hubs have a stronger gathering and radiation power for new media information flow and are able to pay more attention to other cities, as well as receive more attention from other cities compared with cities that are not connected to high-speed railroads.

The influence of mobile phone ownership on dual-perspective centrality is second only to high-speed rail transportation. The contribution of cell phone users per 100 people to the difference of weighted out-degree centrality is 4.29% and passes the 10% significance test. Meanwhile, its contribution to the difference in weighted-in degree centrality is 9.21%, which passes the 5% significance test. The mobile telecommunication industry promotes the development of new media platforms, such as WeChat and Weibo, and increasing their use is an effective way to narrow the information network centrality gap.

The tertiary industry also has an impact on the centrality gap in the outward direction but an insignificant impact on the inward direction. The effect of the contribution rate of the

tertiary industry on the weighted outward centrality difference passes the 10% significance test, but its regression coefficient with the inward centrality difference does not pass the significance test. The development of tertiary industry promotes the strengthening of the city's information radiation power, which makes it receive more attention from outside cities. However, the information agglomeration power, which represents the level of the city's external attention, is not closely related to the tertiary industry.

**Table 1.** Results of QAP regression analysis.

| Factor | Differences of the Weighted Out-Degree Centrality | Differences of the Weighted in-Degree Centrality |
|---|---|---|
| | Standardized Coefficients | Standardized Coefficients |
| Year-end resident population | −0.0362 | −0.0299 |
| GDP per capita | −0.0136 | 0.0549 |
| Contribution rate of tertiary industry | 0.0278 * | 0.0173 |
| Expenditure of public finance | 0.4074 | 0.1981 |
| Number of high-speed trains | 0.6144 *** | 0.6798 *** |
| Number of mobile phone users per 100 people | 0.0429 * | 0.0921 ** |
| Adj R-Sqr | 0.931 | 0.865 |
| Number of replacement | 2000 | 2000 |

Note: ***, ** and * are significant at the statistical level of 1%, 5% and 10%, respectively.

The effects of population, per capita GDP and public financial expenditure on the difference in dual-perspective centrality are not significant. The standardized regression coefficients of the year-end resident population, per capita GDP and public finance expenditure data and the weighted-outward and weighted-inward centrality differences do not pass the significance test. The new media information flow network is not sensitive to the basic strength of the city, and it is the accessibility, including high-speed rail transportation access and telecommunication access, which controls the centrality of the city network.

In general, high-speed railway traffic has the most significant impact on the differences in the perspective centrality of each city in the outbound and inbound directions. The influence of mobile phone ownership and the tertiary industry on the centrality difference is equally significant, and the level of influence is second only to high-speed rail traffic. In contrast, the three indicators of population, GDP per capita and public finance expenditure do not have a significant impact on the differences in centrality.

## 5. Discussion and Conclusions

### 5.1. Discussion

#### 5.1.1. Directional Flow Analysis Further Sub-Divides the Non-Directional Network

Academic studies on flow space networks usually focus on the structural characteristics and organizational patterns of the network [5,10,19,20], which are good for identifying urban systems. However, the practice of constructing element flows is a synthesis of two-way connections between two cities. This method means that it is not possible to subdivide the characteristics of elemental flow in specific directions. Although the interrelation quantity between some cities is very large, this interrelation quantity is internally biased toward one city, and the other city pays significantly less attention to the information of the interlinked cities.

In this paper, we propose a directional and weighted network to study the hierarchical level of bidirectional factor flow between cities. From the analysis results, most directional information flows are clustered at the level of less than 50, and the traffic is generally insufficient. In addition, the information interaction between some cities is seriously

unbalanced. For example, Qianjiang, Zigong, Leshan and other cities have a large amount of information interconnection with the central urban area of Chongqing, but the information flows between these three cities and the central urban area of Chongqing are unequal, with a large amount of information flowing into the central urban area of Chongqing but with significantly lower reverse information flows. This has further extended the research of other scholars. They distinguish the connections between cities in the Chengdu-Chongqing region and determine different levels [35–38]. However, since they consider the total number of connections, a large proportion of advantageous connections do not actually have advantages; on the contrary, they have disadvantages. Because some advantageous links actually rely mainly on one-way information links, the level of information links in the opposite direction is very low. The contrast between the two directions is particularly large, with obvious asymmetry.

5.1.2. Centrality Evaluations Accounting for Status and Direction Information

Cities with higher centrality tend to have a higher level of development because of their central status in cities [47]. The degree of centrality in traditional social network analysis mainly focuses on the analysis of non-directional networks [4,10,20,47]. Some scholars have proposed weighted degree centrality to analyze the importance of differences between cities [45], but this approach targets the degree of importance of nodes integrated in both directions and lacks information about directional splitting. The method also lacks the ability to determine whether a node that occupies an important status in a one-directional perspective would also occupy the same important status in a two-directional framework. Some previous research works have considered the differences in directionality and analyzed the radiation and agglomeration of nodes from the perspectives of both outgoing and incoming directions [49]. However, these studies used an analytical model that emphasizes the connection, judging no direct connection as 0 and a direct connection as 1, and lacks the weighted consideration of the amount of information for specific connections.

This paper builds on these previous studies and incorporates both a directional and weighted approach to analyze the degree of importance of cities in the study area from a two-way perspective. The study finds significant differences in the weighted degree of centrality among cities from a two-way perspective. The results identify the central urban areas of Chengdu and Chongqing as extremely strong nodes. The western parts of sub-cities are identified as strong secondary nodes, and the rest of the nodes have low degree values. Some of the low-intensity cities, such as Dayi, Qianjiang and Nanchuan, are at the bottom level from the outward perspective and belong to the same level as other non-central districts and counties. However, from the inward perspective, these cities are below the level of first- and second-tier cities and also below the level of other non-central districts and counties. This is a good complement to the research of other scholars. They analyzed the status system of cities in non-directional networks through methods such as degree centrality [20,45,47] or analyzed the relationship level of each city in the regional network through the topological association between cities [49]. However, these research works only consider the characteristics under the fixed perspective, and the comparison of characteristics under the specific weight connection perspective of different directions has not been effectively analyzed, so it is impossible to make a comprehensive judgment.

5.1.3. QAP Analysis Was Used to Analyze the Mechanisms Affecting Centrality Differences

Previous research has used QAP analysis to study the factors influencing a variety of correlated network flows, including stock, trade and the resources [50–52]. This method has produced good analytical results. This paper proposes a new method using QAP regression analysis to study the factors influencing the differences in the centrality of cities from a two-way perspective to better understand the coordinated information linkages between cities.

This paper demonstrates that high-speed rail transportation has the strongest influence on the centrality of cities from a two-way perspective. Reducing the gap between cities' high-speed rail transportation development is an important measure to promote the coordinated development of information linkages in the Chengdu-Chongqing Economic Circle. In addition to high-speed rail transportation, mobile phone ownership and tertiary industry also have a strong influence on centrality differences, while population, per capita GDP and public financial expenditure have no significant influence. Overall, the information network has a strong road strength dependence rather than nodal base strength dependence. Smaller cities with better high-speed rail transportation and telecommunication connectivity can obtain stronger centrality in the information flow network.

The analysis results further deepen the research on the influencing factors of the Chengdu-Chongqing Economic Circle information connection network. Previous scholars have explained the important influence of social development on enhancing the urban centrality of the Chengdu-Chongqing Economic Circle through basic methods, such as summary and expert consultation [38]. However, there is a lack of necessary data analysis process for support, so the analysis has certain limitations. This paper discusses the influencing factors through specific data and models and proves the importance of social development, especially informatization and transportation convenience.

### 5.1.4. Thinking and Outlook

This paper is innovative in terms of data, network construction, network analysis and the exploration of influencing factors. Based on the widely used Baidu index data, we propose the correlation data of media articles on new media platforms between cities and construct a directional weighted network based on the data. Different from the commonly used undirected weighted network construction method, the directional weighted network can effectively analyze the asymmetry of information links between cities, rather than being limited to the total number of links between cities. At the same time, we extend the directional thinking to the analysis of network analysis methods and influencing factors and analyze the centrality of each city and the influencing factors of the differences in centrality between cities in different directions. Under this innovative idea, we find serious asymmetric information association between some cities, excavate the urban system of the network and find out the influencing factors of the status difference of each city in the directed information association network. This is a very important contribution to the coordinated development of cities in the Chengdu-Chongqing region, being especially conducive to government decision making and enabling policy makers to put forward the necessary measures to improve the imbalance of information interaction, so as to enhance the information connection between cities, bring the relationship between cities closer and further promote the development of the economy, society and culture. In addition, the ideas and methods of this paper expand the research field of regional network analysis and have special advantages for the analysis of urban interaction in a region, which can be diverse, including traffic interaction, investment interaction, population interaction, innovation interaction and many other aspects. The discussion on the interaction of these different elements in the region will greatly promote the coordination of the development of cities.

The analysis of this paper also has some limitations. On the one hand, since the virtual information data cannot be fully counted, such as the real traffic flow, freight flow and people flow, we refer to the research of other scholars [20,25,39–41] and use the city name association in the new media text to characterize the new media information flow between cities. This has a strong specificity and must be aimed at words that cannot be expanded and specifically refer to cities; otherwise, the amount of information data searched will not be scientific. The names of cities in the study area of this paper meet this characteristic. At the same time, we manually verified the search data in the data collation to avoid the influence of noise articles on the data of this paper. However, when the subsequent scholars apply this method, they need to make a careful judgment instead of directly applying it.

They should carefully check whether the city name satisfies this specificity and make the right choices about the relevant city points in the study. On the other hand, although a regression analysis of the explanatory variables works well, the mechanisms that influence the centrality differences are complex, and it is not possible to accurately quantify many factors, which makes it difficult to analyze comprehensively. Meanwhile, the information flow network does not fully characterize the urban linkages in a real society and only represents one of the multiple linkages between cities. Future research will broaden the research scale and increase the types of research to inject new vitality into the analysis of urban networks.

*5.2. Conclusions*

This paper studies the characteristics of the correlation network of directional and weighted information flow in the Chengdu-Chongqing Economic Circle using new media article data to explore the influence of multiple factors on the differences in city centrality. The results of this study provide an effective theoretical basis for the integration, coordination and sustainable development of the Chengdu-Chongqing Economic Circle.

(1) The central urban areas of Chongqing and Chengdu are the two cores of directional information connection at the intermediate and senior levels in the region, and the quantity of directional information connection between other cities is generally lower than the correlation level between each city and the two cores.

(2) The influence of provincial boundary on information interconnection is quite obvious. The information interaction between sub-cities is very coordinated, but the imbalance of external information interconnection is serious in Chongqing's central urban area, Liangping, Ya 'an and Mianyang.

(3) From a two-way perspective, the centrality of cities in the Chengdu-Chongqing Economic Circle follows a three-tier pattern, with the two centers occupying the first tier, the western sub-cities forming the second tier and the eastern sub-cities and non-central districts and counties in the last tier.

(4) High-speed rail transportation has the most significant influence on the centrality of cities both from the outward and inward perspectives. The influence of mobile phone ownership and tertiary industry on the centrality differences is second only to high-speed rail transportation, while the influence of population, per capita GDP and public financial expenditure is not significant.

**Author Contributions:** Conceptualization, Changwei Xiao, Chunxia Liu and Yuechen Li; methodology, Changwei Xiao and Chunxia Liu; validation, Chunxia Liu, Changwei Xiao and Yuechen Li; formal analysis, Changwei Xiao and Yuechen Li; writing—original draft preparation, Changwei Xiao, Chunxia Liu and Yuechen Li; writing—review and editing, Changwei Xiao, Yuechen Li and Chunxia Liu. All authors have read and agreed to the published version of the manuscript.

**Funding:** This research was funded by the Natural Science Foundation of Chongqing, grant number cstc2019jcyj-msxmX0290, and the Fundamental Research Funds for the Central University, grant number SWU021003.

**Data Availability Statement:** No new data were created or analyzed in this study. Data sharing is not applicable to this article.

**Acknowledgments:** Thanks to all authors for their contributions. Thanks for the funding support.

**Conflicts of Interest:** The authors declare no conflict of interest.

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
