# Peer review of "Directional and Weighted Urban Network Analysis in the Chengdu-Chongqing Economic Circle from the Perspective of New Media Information Flow"

_ijgi, doi:10.3390/ijgi12010001_

Round 1

Reviewer 1 Report

Overall, the paper is very well organized, well-written, and seems ready for publication. I have only a few minor comments that can be easily incorporated.

General comments

C1. The main concern is that the novelty of the research is not fully clear. If such novelty is not clearly highlighted, the risk is that the manuscript looks more a simple case study rather than a research paper.

C2. What are the advantages and disadvantages of this study?

C3. Authors could also emphasize particular strengths of the study for potential applications of their method in other regions.

C4. What are the major contributions of this study? should be carefully mentioned in the discussion section.

C5. Please, look at your Discussion, is there a real comparisons to other researchers of your results?. It is necessary to carry out a thorough comparison, I recommend to include some relevant references, in order to improve the discussion on the novelty of your study, comparing to the others.

*The answer to these questions should be reflected in the manuscript.*

Specific comments

Line 32-33: What does the transfer of urban elements consist of?

Line 34-35: This definition should be supported with relevant references.

Line 38-39: Improve the wording.

Line 45: This is not the correct way to cite. Improve

Line 56-57: What are these studies about "information connection"?

Line 66-69: According to whom. Include relevant references.

Line 88: This is the first time the initials "QAP" are mentioned. Mention the full name.

Line 93-100: Why select Chengdu-Chongqing as the study area? Is it a very special city with specific cultural/natural characteristics in China?

Line 101: Failure to distinguish numbers in Figure 1c. Edit and improve. Also, change "KM" to "km".

Line 102: Use the same font style. Consult the authors' guide.

Line 113: Add the website link of "WeChat" and "Weibo".

Line 121: What are the exploratory variables?

Line 131: In which program was the GIS spatial analysis implemented?

Line 132: In which program was QAP analysis implemented?

Line 154: Change "Where" to "where".

Line 160: Not the correct way to quote, edit.

Line 161: Not the correct way to cite. Revise the author guide and edit.

Line 163: Change "Where" to "where".

Line 175: Change "Where" to "where".

Line 177: Not the correct way to cite. Revise the author guide and edit.

Line 181-184: Sentence is repeated.

Line 192: Change "Where" to "where".

Line 289: Figure is not distinguishable. Improve quality.

Reviewer 2 Report

This paper uses the social media mentions of cities to build a bi-directional graph of relations which are used to  represent information flow between them. The paper is well written and presented, though figures could be improved and do not work well in monochrome. The analytical methods are quite complex, as far as I can tell it makes sense though the nature of the method means it is very hard to tell if it is correct in detail. This said, it is not particularly geographic: the towns have locations and are mapped but the relationships examined are fundamentally topological, the influence of geography is not really considered. Given the focus on social media information that that perhaps makes sense, less so with the other variables such as transportation links, one could argue these act as proxies for geographic friction but one could just as well argue to the contrary.

As to whether conclusions are justified, this really depends on whether one is convinced by the initial assumption that information flow can be measured by using city name as a keyword. If so then I see no reason why the conclusions should not be those reached. Unfortunately, using city names as the sole metric of this is not convincing for two reasons: Firstly there are usually many synonyms, sub locales nicknames etc. which searching for the formal name will miss. Secondly, only some fraction of references in social media will bother to name the geographic place at all e.g. simply stating the name of the stadium, shopping centre etc., assuming its location is understood from that. At the very least some evidence is needed that city name is a reasonable proxy and a more comprehensive keyword set would not materially change the results. However sound the analytical methods thereafter, this early assumption undermines confidence in the conclusion, in my opinion.

Reviewer 3 Report

Dear colleagues, thank you for your interesting contribution on the directional and weighted urban network analysis from the perspective of new media information flow. 

Your approach by using data sources from WeChat and Weibo is a very interesting one. Especially to define regional interconnection, interconnection load, degree of centrality and the QAP regression analysis. 
Although several publications are listed for the influence and data quality of microblogs, there still remains the question for the perspective of the data quality of WeChat and Weibo. How reliant are these data sources? What needs to be considered for a correct analysis?

Some detailed questions and comments exist for this contribution: 

In section 3.3.1. you describe the interconnection load as carrying capacity of vehicles and in the following point to "inter-city connections". What is the carrying capacity of an inter-city connection? The bandwidth? The amount of information? What quality of information? A more clear comparison is needed here. 

In section 3.4.1. you state that you add "directional thinking" with you method. Could you please investigate some words what you mean and how you add directional thinking?

In section 3.4.2 you list the indicators that have been used in the QAP regression analysis. How have you selected these indicators and what is their purpose and quality? There are a lot of more and different indicators that may better fit for purpose. A more precise description on the selection of indicators will be helpful. 

In the analysis you point out that the reverse attention (e.g. line 328) is different. You support this observsation in table 1. Have you considered demographic structure for the population indicators? This might be one possible answer to the reverse attention observation. What is the age structure using WeChat and Weibo? How is the age structure considered in the population indiactors?

The four conclusions of you paper are hardly discussed in your observations. Maybe a clear relation is missing or should be highlighted. 

Thank you for your effort and interesting approach. 

Round 2

Reviewer 2 Report

My apologies for the delay, I was on leave. To expedite matters I will just focus on my primary concern about this paper in the first round, i.e. that it relies on the city name as the key word providing a proxy for interest in the city. The authors have responded by pointing to two references which use this method, one [36] is in Chinese so I cannot read it. The other [25] does indeed apply this method, it is in a good journal and while it has no citations yet that may reflect its recent publication.

However, I do not find the cited paper's use of name as the keyword any more convincing. For example, if one were interested in searching for a football team in Manchester, it is likely one might type "ManU" rather than the full "Manchester United", the latter has of course many more hits but the former has 27 million hits and probably gets the desired information with fewer key presses, yet it would not be picked up by using the formal city name as a keyword. Of course, this may well be language dependent and the city name be less commonly altered in Chinese, but that would mean the method is language specific. Even then, surely there are many other ways to reference interest in a town, to continue the football example e.g. "Old Trafford"?

If the paper had used a larger lexicon, with some reasonable argument for the selected key terms, I would find the paper interesting, the analytical methods look OK.  I remain unconvinced that the town name alone is a sufficient proxy for general interest in something as complex as a city but recognise that other reviewers may disagree.
